# A Comparative Analysis of Measured and Calculated Compressive Stresses of Magnetorheological Fluids under Unidirectional Compression and Constant Area

**DOI:** 10.3390/ma15093057

**Published:** 2022-04-22

**Authors:** Cheng Bi, Hongyun Wang, Wenfei Liu, Keqian Wu

**Affiliations:** College of Aeronautics, Taizhou University, Taizhou 318000, China; bcbicheng@163.com (C.B.); w541503141@163.com (K.W.)

**Keywords:** magnetorheological fluid, compression, unidirectional compression, constant area

## Abstract

Unidirectional compressive properties of magnetorheological (MR) fluids have been investigated under slow compression and constant area with different magnetic fields and different initial gap distances. Experimental tests of unidirectional compression were firstly carried out by using a commercial plate–plate rheometer. The theoretical model based on the continuous squeeze flow theory was developed to calculate the compressive stress. The comparisons between the measured and calculated compressive stresses of MR fluids were made. It showed that the compression resistance of the MR fluid in the magnetic field was much higher than that predicted by the theory. With the increasing magnetic flux density, the deviation between measured and calculated curves accelerated. Characteristics of the compressive stress variation with the reduction in gap distance have been analyzed. The structure strengthening effect induced by the chain structure aggregation in squeeze mode has been used to explain this deviation.

## 1. Introduction

Magnetorheological (MR) fluid is a kind of intelligent material. Its rheological property can be reversibly and instantaneously changed by a magnetic field, similar to electrorheological (ER) fluids [1,2,3]. Extensive research has been carried out to design and control MR devices, such as brakes/clutches, dampers, actuators, etc. [4,5]. Under a zero magnetic field, MR fluids behave much like Newtonian fluids and Bingham fluids under the applied magnetic field [6]. There are three basic working modes for MR fluids in their applications, shear mode, valve mode and squeeze mode. The properties of MR fluids in shear mode have been thoroughly studied [7]. It is widely believed that the shear yield stress of MR fluids is the most important parameter in the design for the shear mode. However, the shear yield stress of MR fluid is so low that it is of no value in the clutches/brakes [8,9]. The experimental research has shown that the squeeze mode can provide ten times higher shear yield stress than when sheared under the same magnetic field [10,11]. The squeeze–strengthen effect is further demonstrated due to the formation of thick columns with strong and robust ends under compression [8]. Therefore, great efforts have been carried out to study the compressive property of MR fluid in the squeeze mode [12,13,14]. The studies found that the initial gap distance, the magnetic field strength and the particle concentration have a significant influence on the compressive performance of MR fluids during the compressive process [7,8,9,10,11,12,13,14,15,16]. The compressive stress is highly dependent and it increases with the increasing magnetic field [7,15,17]. The high magnetic field may increase the attraction between particles and result in the particles forming stronger and more robust structures in the MR fluid [18]. The decreasing initial gap distance during compression has enormous and simultaneous implications for the compressive stress and the shear yield stress of MR fluids [10]. The compressive speed has little influence on the compressive stress when the concentration of MR fluid is low [15].

To predict the compressive resistance of MR fluids in the magnetic field, a continuous media theory based on the Bingham or bi-viscous model has been developed [10]. The relation between the compressive force and the gap thickness has been obtained based on the Bingham mode under the slow squeeze flow [19]. However, the Bingham model was later discovered to be too complex to use in simulations [17]. The bi-viscous mode was found to be appropriate for the presentation of plastic fluids in the slow squeeze flow for ER fluids [20]. A power–law relationship between the compressive force/stress and 1-*ε*, with the exponent −2.5, has been predicted by the continuum squeeze flow theory, where *ε* is the compressive strain [15,16]. In some cases, the validity of the continuum squeeze flow theory can indeed be tested experimentally.

Most of the compression behavior literature reviewed above were often divided into two categories: constant area and constant volume approaches. Constant volume usually refers to plates that were larger than the sample and the radial interface as being free. In constant area geometry, the sample radius was equal to or larger than that of the plates, and the sample flowed out during compression. Here, the tests were performed under constant area conditions. The experimental compressive stress has been found to be underestimated by the continuum squeeze flow theory for ER fluids under the constant area condition, but the validity of this theory for MR fluids has not been verified under the constant area operation.

Although the validity of the continuum squeeze flow theory has been proven to underestimate the experimental compressive stress for ER fluids under the constant area condition [21,22], it has not been proven for MR fluids. Few studies have been published which consider whether this theory is suitable to manage the unidirectional compression under constant volume [15,16]. Guo et al. found that the power law indices agree with the theoretical analysis of continuous media theory only when the concentrations of MR fluids are between 10% and 25% [15]. The index deviates from the theoretical value when the concentration reaches 30%. José et al. discovered the deviations of the curves of normal force versus compressive strain at small strains under constant volume operation [16]. The similar deviations have also been found in the compression study of ER fluids under constant area [21,22,23]. They found that this theory underestimates the compressive stress at small initial gaps and high voltages. Therefore, there is still a lack of detailed experimental and theoretical research on the compression behavior of MR fluids under constant area operation. Moreover, in order to better understand the compressive mechanism of MR effect, it is essential to investigate the relationship between the compression behavior and chain structure aggregation.

In this study, the unidirectional compressive experiments under different magnetic field and initial gap distances at a low compressive speed were conducted under constant area by using a commercial plate–plate rheometer. The theoretical model, based on the continuous media theory, was first described in detail and then carried out to calculate the compressive stress. The experimental results under different magnetic fields and initial gap distances were presented. The comparison between the measured and calculated compressive stresses of MR fluids was made. Deviations from the continuous squeeze flow theory were discovered and attributed to the structure strengthening effect induced by the chain structure aggregations in squeeze mode.

## 2. Experimental Details

All analyses of the squeeze flow behavior of MR fluids were conducted using an Anton Paar rheometer (Model: MCR302) with Ti plates with a rim to use with their commercial magnetocells. It has a diameter of 20 mm for both plates. The maximum of the measurable normal force is 50 N. The magnetic field is generated by the coils. The range of applied current generated by the coils is 0~5 A. In this research, the MR fluid of MRF-2035 from Ningbo Shangong Co. Ltd., Ningbo, China, was employed in this experiment. It is based on dimethyl silicon oil and iron powder. It has the particle volume fraction of 35%.

To obtain the shear yield stress, the shear stress versus shear rate of the MR fluid was tested by this rheometer under different applied magnetic fields. To obtain the compressive stress, two sets of experiments were conducted. At first, the initial gap distance *h*_0_ = 1 mm and different current values (1, 2, 3 and 4 A, corresponding to 0.28, 0.45, 0.63 and 0.81 T, respectively) were applied. Then, the upper plate moved downward at a constant compressive speed *v* to compress the MR fluid. Between the constant normal force applied or the constant compressive speed, the constant compressive speed was chosen in these experiments at a displacement control mode. To guarantee the low *S* numbers and the so-called filtration regime of operation, the low compressive speed *v* = 50 μm/s was used. During compression, the applied current was kept constant and was withdrawn after the compression. The same procedure was carried out in the second set of experiments, except for the initial gap distance *h*_0_ = 2 mm. For instance, when *h*_0_ = 2 mm, the applied currents are 1, 2, 3, and 4 A, respectively. Moreover, the compressive speed is kept constant i.e., *v* = 50 μm/s.

The compressive stress can be calculated as *P* = *F*/*S*, where *F* is the compressive force and *S* is the area of the plate. The compressive strain can be expressed as *ε* = (*h_0_* − *h*)/*h*_0_, where *h* is the instantaneous gap distance between the two plates. The magnetic flux density versus the applied current is shown in Figure 1. All experiments were performed at room temperature.

## 3. Theoretical Analysis

Under squeeze flow, MR fluids are compressed between two parallel plates with a radius *r* and a gap distance *h*, as shown in Figure 2. The upper plate moves slowly at the speed *v* toward the static bottom plate to compress the MR fluid. According to the squeeze flow theory by Williams et al. using the Bi-viscosity model instead of the Bingham model [24,25], the compressive stress can be represented as:(1)P=2τyr0hβ3[k3108+∫k3βGWdG]
where *τ_y_* is the dynamic yield shear stress, which is taken as a form for *τ_y_ = K*(*B*)*^m^* (*K* and *m* are the constants related to the material); *r*_0_ is the radius of the upper plate; *k* (the coefficient of viscosity) is the ratio of *η_H_* and *η* (*η* and *η_H_* are the pre-yield and post-yield viscosity in the Bi-viscous model, respectively), *k* is the value of 10^−5^~10^−2^ and *τ_y_ = τ*_0_ (1 − *k*), *τ*_0_ is the static yield shear stress; *G, β* and *W* are the parameters defined, respectively, by:(2)$$G=\frac{\eta vr}{h^{2}\tau_{y}};\;\beta ={G|}_{r=r_{0}};\;W=-\frac{h}{2\tau_{y}}\frac{dN}{dr},$$
where *G* is a plasticity number, *W* is a dimensionless pressure gradient, and it is often defined as:(3)$$W^{3}-3\left(G+\frac{1}{2}\right)W^{2}+\frac{1}{2}=k\left(\frac{1}{2}-\frac{3}{2}W^{2}\right)$$

Taking *k* = 10^−4^ << 1, Equation (3) can be simplified as:(4)$$W=1+\sqrt{2G}$$

Substituting Equation (4) into Equation (2) and combining with Equation (1), the compressive stress can be obtained as:(5)$$P=\frac{2r_{0}}{3h}\tau_{y}+\frac{4r_{0}}{7h^{3}}\tau_{y}\sqrt{2G}$$

The curves of shear stress versus shear rate is measured using the rheometer MCR 302 at different applied currents, as shown in Figure 3a. The shear yield stress with magnetic field is fitted using the Bingham model as *τ_y_* = 10,859*B*^1.53^, as shown in Figure 3b. Taking *r*_0_ = 20 mm, *η =* 0.24 Pa/s, *v =* 100 μm/s and *τ_y_* = 10.9*B*^1.53^ (the units of *τ_y_* and *B* are Pa and mT, respectively), the compressive stress can be calculated by Equation (5) under different magnetic flux density *B.* For example, *P* = 21,054/*h* + 0.23/*h*^3^ when *B* = 0.45 T and *P* = 52,316/*h* + 0.57/*h*^3^ when *B* = 0.81 T. The plasticity number *G* is always found to be smaller than 0.03, which is agrees with the result for *G <* 0.05 by Vicente et al. [10]. The second term in Equation (5), in this case, is always found to be less than 10 because *G <* 0.03. Under this condition, the viscous effect has little contribution to the compressive stress *P* and may be negligible. Therefore, when *k* ≈ 0 (namely, MR fluids are Bingham fluid), Equation (5) can be simplified as:(6)$$P=\frac{2r_{0}}{3h}\tau_{y}$$

When *k* = 1 (namely, MR fluids are Newton fluids), taking *τ_y_ = τ*_0_ (1 − *k*), MR fluids degenerate into a pure squeeze flow of Newtonian fluids. Hence, Equation (5) can be simplified as:(7)$$P=\frac{3\eta r_{0}^{2}v}{2h^{3}}$$

At this time, it is pure squeeze flow of Newtonian fluid and can no longer reflect the bearing of a fluid with yield stress; therefore, it does not belong to in the scope of this study.

The shear yield stress of MR fluid is a function of the magnetic field and can be calculated by *τ_y_* = *K**B*^m^ (*m* = 1.5 for moderate fields and *m* = 2 for small magnetic fields). The study also found that the shear yield stress *τ_y_* of MR fluids is proportional to the instantaneous gap distance *h* with an exponent as *τ_y_* = *A*/*h*^m^ (*A* and *m* is constant, *m* = 1.5~2) [26]. By introducing the relationship of *τ_y_* = *A*/*h*^m^, Equation (6) can be rewritten as:(8)$$P=\frac{2Ar_{0}}{3h^{m+1}}$$

When the instantaneous gap distance *h* takes from the initial gap distance *h*_0_, the special compressive stress *P*_0_ can be written as:(9)$$P_{0}=\frac{2Ar_{0}}{3h_{0}^{m+1}}$$

Dividing Equation (8) by Equation (9), the natural logarithmic form can be obtained:

ln*P* = *W –* (*m* + 1) ln*h*
(10)

where *W* = ln*P*_0_ + (*m* + 1) ln*h*_0_. Because the slope of (*m* + 1) is a constant for a given MR fluid, ln*P*_0_ plus 0.69(*m* + 1) is constant when *h*_0_ is fixed. Equation (10) means that *W* is the intercept and (*m* + 1) is the slope in the linear function between ln*P* and ln*h*. It shows that in the ln–ln plot, ln*P* varies linearly with ln*h* with a constant slope of –(*m* + 1), no matter what the magnetic flux density, the initial gap distance and the material property factor *A*.

## 4. Results and Discussion

For each initial gap distance *h*_0_ (1 and 2 mm), *B*_0_ were set to 0.28, 0.45, 0.63 and 0.81 T, respectively. The typical results of the compressive stress *P* versus the instantaneous gap distance *h* in squeeze mode are shown in Figure 4 when the initial gap distances are *h*_0_ = 1 mm and *h*_0_ = 2 mm. It shows that the curves of compression can be roughly divided into two regions, as shown in Figure 4a. In the first region, the compressive stress increases sharply with the decrease in the instantaneous gap distance and a dramatic jump-like increasing behavior can be observed in the curve. This first region may be called the elastic deformation region, where the structure of chains is formed at the applied magnetic field and becomes denser and denser without breaking during compression. In the second region, the instantaneous gap distance is lower than 0.974 and 1.948 when *h*_0_ = 1 mm and *h*_0_ = 2 mm, respectively. The compressive stress increases with the decrease in the instantaneous gap distance. This second region can be called the plastic flow region, where the structure of chains collapses, reorganizes and forms stronger and more robust column/body-centered cubic (BCC) structures with the decreasing distance. Additionally, *P* at a higher applied magnetic field is distinctly higher than that at a lower applied magnetic field when *h* is certain.

The larger experimental values of compressive stresses are obtained when *h*_0_ =1.0 mm in comparison to the situation when *h*_0_ =2.0 mm has been set under *B*_0_ = 0.45 T and *B*_0_ = 0.63 T, as shown in Figure 5. For instance, *P* is approximately 188 kPa and 146 kPa when *h*_0_ = 1 mm and *h*_0_ = 2 mm, respectively, at *ε* = 0.091 and *B =* 0.63 T. It reveals that the compressive stress of MR fluid is strongly affected by the initial gap distance, and a smaller initial gap distance can generate a larger compressive stress when the compressive strain is the same. This result is contrary to the result of Mazlan et al. [7], which indicates that the compressive stress at a smaller initial gap distance is lower than that at a larger initial gap distance. However, it agrees with the result for MR fluids [15] and ER fluids [21].

The MR fluid was tested on the rheometer under different conditions of initial gap distance and applied magnetic field, as listed in Table 1. The theoretical compressive stresses, calculated from Equation (5), and the experimental compressive stresses (*P* = *F*/*S*) at *ε* = 0.2 for each test are listed in the last column in Table 1. The comparison of compressive stresses between the measured and calculated results is shown in Figure 6 under *B*_0_ = 0.45 T when *h*_0_ =1.0 mm and *h*_0_ =2.0 mm. It shows that the calculated and experimental compressive stresses do not agree well. Namely, the calculated compressive stresses are smaller than the experimental ones. It means that the theoretical value underestimates the compressive stress. Moreover, this deviation increases with the decrease in the instantaneous gap distance or the increase in compressive strain. Similar results have also been obtained by Guo et al. concerning the compressive performance of MR fluids under a nonuniform field [26].

Taking the logarithm of *P* and *h* values in Figure 4a,b, the experimental curves of ln*P* versus ln*h* are shown in Figure 7a,b, respectively. The prediction lines calculated according to Equation (10) are also shown in Figure 7 with *m* = 1.53. When *h*_0_ = 1 mm, fitting the curves in Figure 7a with the linear functions, the equations for the curves are ln*P* = 2.74–3.01 ln*h* (0.28 T), ln*P* = 4.19–3.21 ln*h* (0.45 T), ln*P* = 4.91–3.37 ln*h* (0.63 T) and ln*P* = 5.17–4.24 ln*h* (0.81 T). The experimental slope no longer agrees with the theoretical one and it is greater than that of 2.53, as shown in Figure 7a. The average value of the slopes is 3.46. It shows the trend that the slopes (*m* + 1) are obviously different from each other and increase with the increase in the magnetic flux density. The traditional dipole model of MR fluid predicts a square dependence on the shear yield stress with the magnetic field, namely, *m* = 2. That means that the prediction of the squeeze flow theory or the traditional dipole model both underestimate the compression resistance of MR fluid, no matter if m = 1.53 or m = 2. This phenomenon is also observed in other compressions of ER fluids [21].

With the further increase in *h*_0_ to 2 mm, the experimental slopes vary from 2.68 to 3.46 with the increase in the magnetic flux density and their average value is 3.05, as shown in Figure 7b. Obviously, the slope gets small as the initial gap distance increases. Most of the test results of the slope are higher than 2.53 at *h*_0_ = 2 mm. It implies that the compression resistance of the MR fluid increases more than the theoretical prediction with the decrease in the instantaneous gap distance. These experimental results also indicate that the description of compressive behavior of MR fluids with the squeeze flow theory might not reveal the essential attributes of the MR effect during compression. The difference between the theory and experiment at a smaller *h*_0_ is obviously greater than that at a larger *h*_0_ when the magnetic flux density is the same, and it is also greater for the higher applied magnetic field than the lower when *h*_0_ is the same. This suggests the failure of the traditional description of the compression behavior of MR fluids based on the squeeze flow theory, especially in the case of small initial gap distance and high applied magnetic field. This result agrees with the result for ER fluids by Tian et al. [21].

MR fluids are suspensions of magnetic micro-particles in a carrier fluid. The particles become polarized and interact strongly with each other, forming chains along the direction of the magnetic field under an applied magnetic field. To explain the compressive behavior of MR fluid, the mechanical property of MR fluids under compression needs to consider the evolution of particle aggregation. 

The compressive resistance is primarily contributed by the chain resistances from the field-induced yield stress [15]. The particle chains in MR fluids can be described by using a slim rods model. According to the mechanics of compressed slim rods, the rod strength (*P_g_*) can be expressed by the following equation [16,21]:(11)$$P_{g}=fQ^{2}$$
where *f* is a material parameter that is called the structure factor; *Q* is the aspect ratio *d/h_g_,* where *d* is the rod diameter, *h_g_* is the rod length that is equal to the instantaneous gap distance *h* of the compressed MR fluid. When the magnetic field is applied, the particles in MR fluids form the chain structures along the direction of the magnetic field in milliseconds, assuming that the rod/chain diameter and the rod/chain length are *d* and *h_g_*, respectively, before compressed. The particle chains collapse, reorganize and form stronger and more robust column/BCC structures with the decreasing gap distance during compression. Coarse columns composed of several chains and BCC structures have been observed through microscopy [15,18]. Such structures have the effect of enlarging the rod/chain diameter *d.* Therefore, the rod/chain length *h_g_* decreases and the rod/chain diameter *d* increases during compression, which means the increasing of the aspect ratio *d*/*h_g_*. The increasing aspect ratio *Q* during compression leads to the increase in the rod strength/chain resistance *P_g_*, which is similar to the squeeze–strengthen effect reported by Zhang et al. [8]. For the different initial gap distance, the aspect ratio *Q* is mainly dependent on *h*_0_ at the beginning of the compression. The longer the rod/chain length *h_g_*/*h*_0_, the more likely the chain structures are to collapse during compression. The rod/chain diameter *d* is constant at the same magnetic flux density and the aspect ratio *Q* at a smaller *h*_0_ is clearly larger than that at a larger *h*_0_ at the beginning of the compression. Thus, the compressive resistance of MR fluid at *h*_0_ = 1 mm is larger than that at *h*_0_ = 2 mm, as shown in Figure 5 and Figure 7. Meanwhile, the more robust structures formed by compression increase the material structure parameter *f* too, as reported by Tao et al. Therefore, the change in microstructure leads to the greater increase in compressive resistance. However, the model, as assumed in the William’s squeeze-flow theory, is built based on single chains and it is suitable for the shear mode, but it is not always applicable in squeeze mode. The compressive stress calculated from Equation (5) is much lower than the experimental, especially in the case of small initial gap distance and high applied magnetic field, as shown in Figure 6. The particle interactions mechanism in the microstructure of MR fluids under compression is attractive and further research needs to be conducted.

## 5. Conclusions

In this study, the squeeze behaviors of MR fluids have been studied under different magnetic fields and initial gap distances in slow compression. The experimental results have revealed that compressions under different magnetic fields can always make MR fluids strong. The compressive stresses have been calculated according to the theoretical model based on the continuous squeeze flow theory. Experimental results of compressive stress were compared with the calculated values. The difference between the experimental and the theoretical curves significantly increases with the increasing magnetic field, which seems to deviate from the prediction based on the continuous squeeze flow theory. Characteristics of the compressive stress variation with the reduction in gap distance have been analyzed. The structure strengthening effect caused by the chain structure aggregation is currently utilized to account for this deviation in the compression of MR fluids, which may well be referred to for use in quantitative research for MR strength.

## Figures and Tables

**Figure 1 materials-15-03057-f001:**
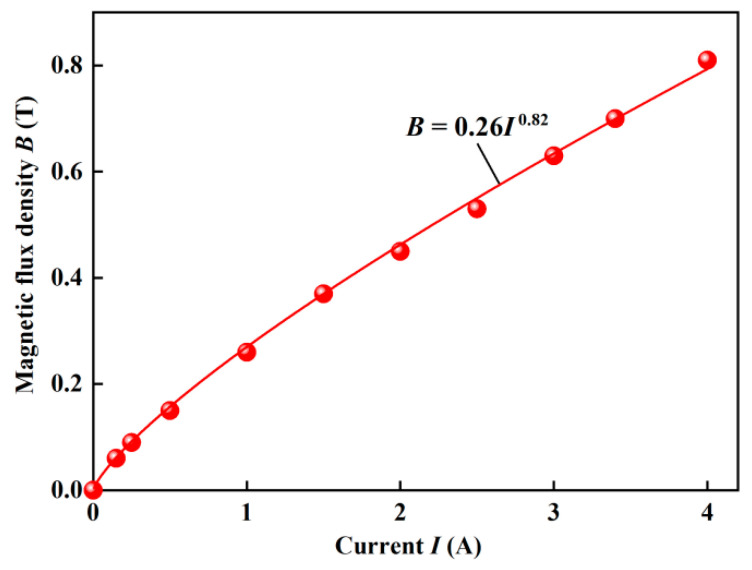
The magnetic flux density versus the applied current.

**Figure 2 materials-15-03057-f002:**
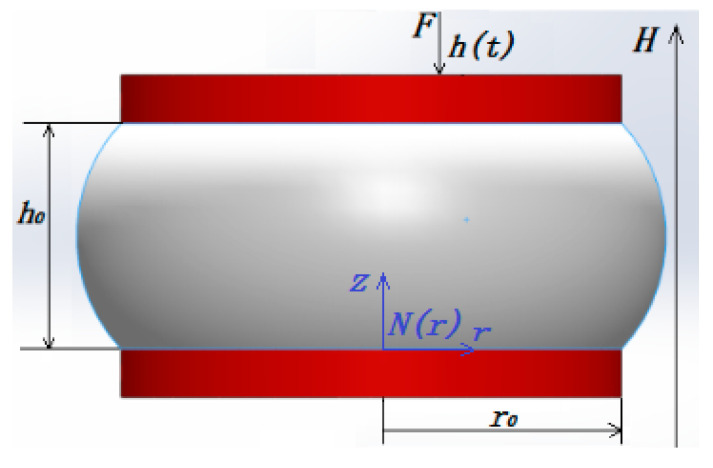
The sketch of the compression of MR fluids between two parallel plates.

**Figure 3 materials-15-03057-f003:**
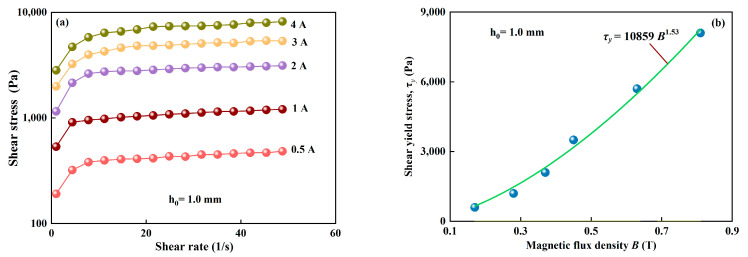
Mechanical properties of the MR fluid under shearing. (**a**) Shearing curves of shear stress versus shear rate at different applied currents in a range from 0.5–4A, corresponding to 0.17–0.81 T; (**b**) the shear yield stress versus magnetic fields.

**Figure 4 materials-15-03057-f004:**
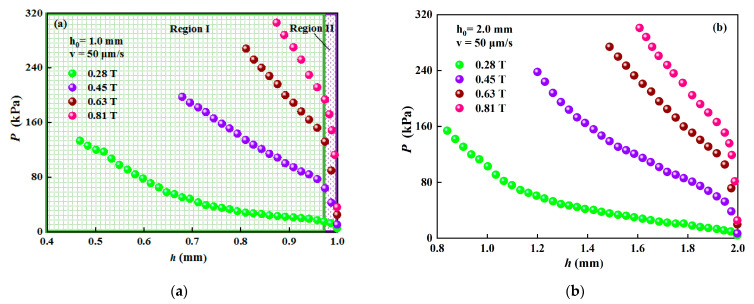
The compressive stress versus the instantaneous gap distance under different magnetic fields at v = 50 μm/s. (**a**) *h*_0_ = 1 mm; (**b**) *h*_0_ = 2 mm.

**Figure 5 materials-15-03057-f005:**
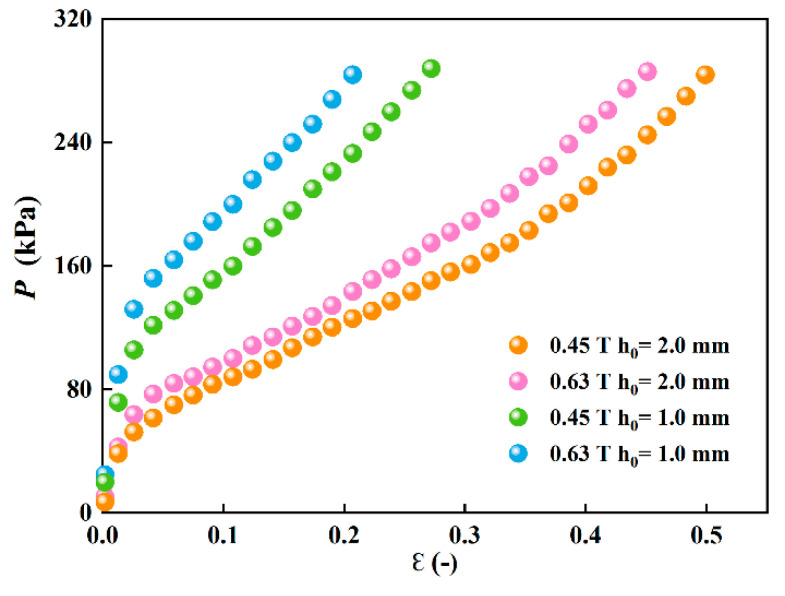
Comparison for the value of compressive stresses between two different initial gap distances (*h*_0_ = 1 mm and *h*_0_ = 2 mm) under two magnetic fields.

**Figure 6 materials-15-03057-f006:**
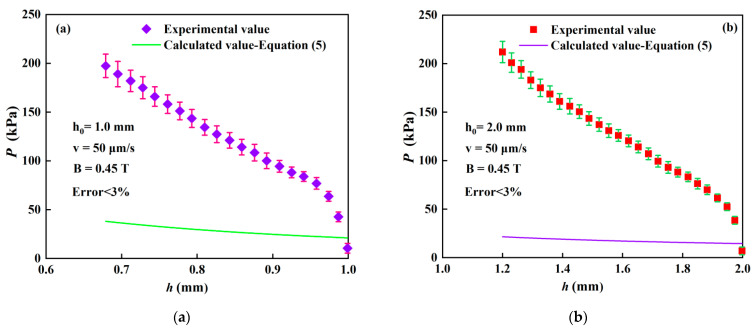
Comparison of compressive stresses between the measured and calculated results. (**a**) *h*_0_ = 1 mm; (**b**) *h*_0_ = 2 mm.

**Figure 7 materials-15-03057-f007:**
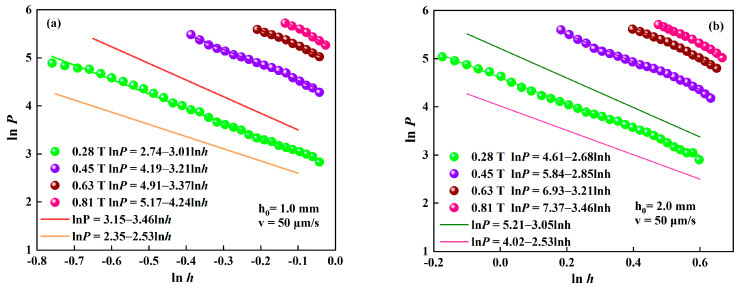
Logarithmic curves of compressive stress versus the instantaneous gap distance with different magnetic fields at v = 50 μm/s. (**a**) *h*_0_ = 1 mm; (**b**) *h*_0_ = 2 mm.

**Table 1 materials-15-03057-t001:** Test conditions and the calculated and experimental values of compressive stress.

Initial Gap Distance (mm)	Magnetic Field (mT)	Experimental *P* (kPa) at ε = 0.2	Theoretical *P* (kPa) at ε = 0.2
*h*_0_ = 1	0.28	23	12.84
*h*_0_ = 1	0.45	108	25.98
*h*_0_ = 1	0.63	216	44.16
*h*_0_ = 1	0.81	306	64.89
*h*_0_ = 2	0.28	21	6.42
*h*_0_ = 2	0.45	93	12.99
*h*_0_ = 2	0.63	172	22.08
*h*_0_ = 2	0.81	230	32.47

## Data Availability

Not applicable.

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
