# Peer review of "A Comparative Analysis of Measured and Calculated Compressive Stresses of Magnetorheological Fluids under Unidirectional Compression and Constant Area"

_materials, 2022, doi:10.3390/ma15093057_

Round 1
Reviewer 1 Report
Magnetorheological (MRF) fluids have been investigated in squeeze mode. Slow compression velocity and constant area with different magnetic fields and initial gap distances were considered. The tests were carried out on a rotary rheometer (MCR302). The main aim of the study was to compare the results of the experiment with the compressive stress value based on theory by Williams. This formula was used in the case of ER fluids, and in the case of MRF for the particle volume fraction in the range of 10-25%. The study took into account the MRF-2035 fluid with a particle volume fraction of 35%. A significant difference between the calculations and the experiments was shown. The main reason for that was indicated that the model does not include forming chains along the direction of the magnetic field (slim rods model). Results are presented clearly and analysed appropriately, but the results are not innovative.
My Comment:
1. Why was the gap size 1 mm and 2 mm chosen ?. Was there any fluid spilling sideways at the 2mm gap?
2. Similarly, why was the speed v = 50 µm/s chosen ? Would similar resutalts come out with others values?
3. figure 2, Figure 1, figure 3 (a) - One time is lowercase, another time is uppercase.
4. In the introduction, the velocity v = 50 µm/s is written. Below the figures there is a value of v = 100 μm/s
5. A rheometer model is written one time 301 another 302.
Reviewer 2 Report
Report
- Unidirectional compressive properties of magnetorheological (MR) fluids have been investigated under slow compression and constant area with different magnetic fields and initial gap distances.
- Experimental investigation is done by using a commercial plate–plate rheometer.
- The theoretical model based on the continuous squeeze flow theory which is used to calculate the compressive stress.
- The compression resistance of the MR fluid in the magnetic field is much higher than that predicted by the theory.
- Increasing magnetic flux density, the deviation between measured and calculated curves accelerates.
- All analyses were carried out using an Anton Paar rheometer (Model: MCR302) with Ti plates with a rim to use with their commercial magneto cells.
- The magnetic field is generated by the coils.
- The range of applied current generated by the coils is 0~5 A.
- The structure strengthening effect caused by the chain structure aggregation is currently utilized into account for this deviation on the compression of MR fluids.
- Overall, the article is organized well and suggested to publish with some more details about the methodology of obtaining results.
